# SEMI-SUPERVISED OFFLINE REINFORCEMENT LEARNING WITH PRE-TRAINED DECISION TRANSFORMERS

## ABSTRACT

Pre-training deep neural network models using large unlabelled datasets followed by fine-tuning them on small task-specific datasets has emerged as a dominant paradigm in natural language processing (NLP) and computer vision (CV). Despite the widespread success, such a paradigm has remained atypical in reinforcement learning (RL). In this paper, we investigate how we can leverage large reward-free (i.e. task-agnostic) offline datasets of prior interactions to pre-train agents that can then be fine-tuned using a small reward-annotated dataset. To this end, we present Pre-trained Decision Transformer (PDT), a simple yet powerful algorithm for semi-supervised Offline RL. By masking reward tokens during pre-training, the transformer learns to autoregressivley predict actions based on previous state and action context and effectively extracts behaviors present in the dataset. During fine-tuning, rewards are un-masked and the agent learns the set of skills that should be invoked for the desired behavior as per the reward function. We demonstrate the efficacy of this simple and flexible approach on tasks from the D4RL benchmark with limited reward annotations.

## 1 INTRODUCTION

Over the past decade, we have seen tremendous progress in artificial intelligence, due in large part to our ability to train deep neural network models using massive amounts of data. The initial success of training such deep models was achieved by using large *labeled* datasets (e.g. ImageNet (Deng et al., 2009)), which provided rich and direct supervision for various tasks (e.g. in CV and NLP). However, the need for human annotations limited the scalability of such an approach, especially for tasks requiring detailed annotations like instance segmentation or machine translation (Lin et al., 2014; Bojar et al., 2016). Self-supervision has emerged as a dominant paradigm to overcome this limitation, where deep models are pre-trained using large unlabeled datasets, followed by fine-tuning with a small labelled dataset, considerably reducing the annotation requirements (Chen et al., 2020; He et al., 2020; Grill et al., 2020).

Analogously, progress in RL has required human annotations in the form of detailed reward functions. While some exceptions exist where the environment computes the rewards (e.g. games like Atari and Go); in most real-world applications of RL, a human is required to design reward functions or annotate frames for rewards. Such well-engineered reward functions are crucial for an agent's ability to learn successful policies, which is hard to scale. At the same time, collecting reward-free and task-agnostic datasets for RL is becoming increasingly easier through use of offline datasets (Fu et al., 2020), tele-operation and play datasets (Rajeswaran et al., 2018; Lynch et al., 2019), or reward-free exploration (Pathak et al., 2017; Eysenbach et al., 2018; Liu & Abbeel, 2021). In this work, we aim to design RL agents that can utilize large reward-free datasets for unsupervised pre-training as well as small reward-annotated datasets for supervised finetuning to solve a variety of downstream tasks.

Figure 1: (left) The pre-training phase: A transformer predicts actions conditioned on states and rewards that are masked and is trained on the un-annotated large offline dataset. (right) The fine-tuning phase where a small fraction of the data is reward-labeled, we unmask the reward tokens and predict the reward as well as actions during fine-tuning on the downstream task.

To this end, we consider the problem of semi-supervised offline RL, which is conducive for investigating our research question. In this setting, the agent has access to a large reward-free (task-agnostic) offline dataset of environment interactions. Such datasets are easier to scale than reward-labeled data and can be re-used for several downstream tasks with different rewards, similar to the trend of pre-training on unlabeled data and adapting to supervised downstream tasks in CV (Dosovitskiy et al., 2020) and NLP (Devlin et al., 2018; Brown et al., 2020). The agent further has access to a small dataset containing reward annotations, which anchors the agent to learn the desired behavior or task.

For this setting, we introduce a new approach: Pre-trained Decision Transformers (PDT), an architectural extension to the recently introduced Decision Transformer (DT; Chen et al. (2021)), to make it amenable to semi-supervised (offline) RL. DT frames RL as a sequence modeling task using an autoregressive and causally masked transformer model, and was shown to be competitive with traditional RL algorithms based on temporal difference learning. As shown in Figure 1, our framework extends DT and consists of two steps: *pre-training* and *fine-tuning*. First, from a large and fixed offline data, we use the state and action context of trajectories to predict the next actions autoregressively with the DT architecture. To accommodate fine-tuning with reward signals, during this pre-training step, we mask the reward tokens by zeroing them out as well as modifying the causal mask of the self-attention layers to generate actions without reward tokens. This can enable the model to implicitly learn the skills present in the offline un-labeled dataset. Then in the fine-tuning step, we un-mask the reward tokens and fine-tune the transformer end-to-end, predicting both reward and action tokens. This allows the agent to use the information present in the small reward-annotated dataset to adapt its set of skills towards achieving high rewards in the downstream task.

We summarize our contributions as follows:

1. We propose a new *semi-supervised offline RL* setting to bring RL closer to the self-supervised learning paradigm in CV and NLP.

2. We introduce Pre-trained Decision Transformers (PDT) - an extension of the recently proposed Decision Transformers which masks the reward tokens during pre-training and can automatically generate reward-to-gos during evaluation.

3. We show how PDT can efficiently be fine-tuned for a range of diverse downstream tasks using the same pre-trained model in a few-shot manner, achieving leading aggregate performance on D4RL environments in the low-label regime.

4. We provide insight into how different parts of PDT contribute to the final performance.

## 2 RELATED WORK

**Behavioral Cloning (BC).** When a reward-free but expert-quality dataset is available, imitation learning is often considered as a simple and yet powerful baseline for learning effective policies Osa et al. (2018); Rahmatizadeh et al. (2018). Our work differs in the motivation to utilize sub-optimal reward-free experience in the pre-training phase of the agent.

**Offline RL.** Offline RL is a paradigm for RL where agents are trained on offline datasets rather than directly interacting with the environment. The main issue in utilizing existing value-based off-policy RL algorithms in the offline setting is that these methods often erroneously produce optimistic value function estimates that can encourage the policy to pick actions that are out-of-distribution (OOD) compared to the offline dataset, resulting in failure. One way to mitigate this problem is to constraint the learned policy to be "close" to the data generating policy (also known as behavior policy) , by means of KL-divergence (Jaques et al., 2019; Wu et al., 2019a; Siegel et al., 2020; Peng et al., 2019a), Wasserstein distance (Wu et al., 2019a), or MMD (Kumar et al., 2019), and then using the sampled actions from this constrained policy in Bellman backup or applying a value penalty based on these divergence measures. Another way is to be conservative on value estimation by adding penalty terms that enforces the models to learn a lower bound on the value functions (Kumar et al., 2020; Fujimoto et al., 2019). All these methods however require the offline datasets to be reward-annotated, and rely crucially on rewards for learning.

Another line of work extends Model Based Reinforcement Learning (MBRL) methods to offline settings. Similar to model-free methods, existing MBRL methods cannot be readily used in offline settings due to the distribution shift problem and require some safeguard to mitigate exploiting model inaccuracies. To mitigate this issue, Kidambi et al. (2020) and Yu et al. (2020) both incorporate pessimism in learning the dynamics and estimating uncertainties in reward learning. While the dynamics models themselves can be learned without reward functions, planning or policy learning using the learned model again relies crucially on having access to a reward function or a large reward-annotated dataset.

**Skill Extraction with Behavioral Priors.** Methods that leverage behavioral priors utilize reward-free environment interactions to learn the set of task-agnostic skills via either maximizing likelihood estimates (Pertsch et al., 2020; Ajay et al., 2020; Singh et al., 2020) or maximizing mutual information (Eysenbach et al., 2018; Sharma et al., 2019; Campos et al., 2020). Behavioral priors learned through maximum likelihood latent variable models have been used for structured exploration in RL (Singh et al., 2020), to solve complex long-horizon tasks from sparse rewards (Pertsch et al., 2020), and regularize offline RL policies (Ajay et al., 2020; Wu et al., 2019b; Peng et al., 2019b; Nair et al., 2020). However, these skill extraction algorithms have high architectural complexity, requiring a generative model to learn a latent skill space and an RL agent to learn policies over skills as its action space. In contrast, PDT has one simple architecture for both skill extraction by modeling state-action sequences and control by conditioning on the desired reward.

## 3 APPROACH

### 3.1 PRELIMINARIES

**Offline Reinforcement Learning.** We consider the standard RL setting of a Markov Decision Process (MDP) defined by the tuple $\mathcal{M} = (\mathcal{S}, \mathcal{A}, \mathcal{R}, \mathcal{T})$. The MDP tuple consists of state $s \in \mathcal{S}$, actions $a \in \mathcal{A}$, rewards $r = \mathcal{R}(s, a)$, and transition dynamics $\mathcal{T}(s'|s, a)$. A policy in RL aims to pick actions that will maximize the expected total return of the MDP $\mathbb{E}\left[\sum_{t=0}^{T} \mathcal{R}(s_t, a_t)\right]$. One episode can be represented as a trajectory in the MDP consisting of states, actions and rewards $\tau = (s_1, a_1, r_1, s_2, a_2, r_2, ..., s_T, a_T, r_T)$.

Offline RL is a setting of RL that seeks to find the optimal policy from a fixed offline dataset of interactions. $\mathcal{D} = \{s_i, a_i, r_i\}_{i=1}^{N}$. As a result, this setting is more difficult than the standard RL setting due to lack of access to the environment, and therefore is prone to distribution shift.

**Decision Transformers.** The usage of transformers (Vaswani et al., 2017) to tackle reinforcement learning problems has been a topic of recent study in Chen et al. (2021) and Janner et al. (2021). In Decision Transformer, rather than learning explicit policies or value functions, trajectories are modeled as sequences of state, actions, and reward-to-go. A reward-to-go (RTG) is defined as the sum of future rewards $\hat{R}_t = \sum_{t'=t}^{T} r_{t'}$. Then the problem setting becomes a sequence modeling problem, where actions are generated conditioned on the past context, current state, and the desired reward-to-go. With a context of length $H$ at current time step $t$ (i.e. $C_t^{(H)}$), the policy becomes:

$$p(a_t | s_t, \underbrace{\hat{R}_{t-1}, a_{t-1}, s_{t-1}, ... \hat{R}_{t-H}, a_{t-H}, s_{t-H}}_{C_t^{(H)}})$$

The architecture of Decision Transformer first applies a linear token embedding to states, actions, and RTGs, and then adds a learned positional embbedding (shared across tokens within the same time-step) to differentiate between similar tokens at different positions along the sequence. It uses a GPT (Radford et al., 2019) architecture to learn a policy that auto-regressively generates actions based on prior context. The loss during training DT is a mean-squared error loss between predicted actions and ground truth actions.

### 3.2 OFFLINE RL WITH PRE-TRAINED DECISION TRANSFORMERS

Our approach is composed of two stages: Pre-training and Fine-tuning. During pre-training the model learns to extract the common temporally extended sequences of actions from the pre-training dataset. Our hope is that by fine-tuning this model on the downstream reward-annotated dataset it will be able to quickly learn task-specific knowledge including a reward prediction model and a reward-conditioned policy. We now elaborate each part in more details:

#### 3.2.1 PRE-TRAINING DT

During pre-training, a sequence of trajectory is modeled as:

$$\underbrace{[s_1], [a_1], [RTG]}_{\text{Pos.1}}, \underbrace{[s_2], [a_2], [RTG]}_{\text{Pos.2}}, ..., \underbrace{[s_T], [a_T], [RTG]}_{\text{Pos.}T}$$

Where [RTG] is the placeholder token for reward-to-go which is hard-coded to zero during pre-training. Note that the same positional encoding is added to state, RTG, and action of the same time-step. We also modify the causal mask in the self-attention layer to disregard any attention paid to the [RTG] tokens. This is crucial for the model to be reward-agnostic, and therefore, allowing for a quicker adaptation to downstream RTGs fed in at fine-tuning time. Pre-training is done by computing the maximum-likelihood estimates (MLE) of the neural network parameters.

$$\mathcal{L}_{PT}(\theta) := \mathbb{E}_{(s_t, a_t, C_t) \sim \mathcal{D}^{PT}} \left[ -\log p_\theta \left( a_t | s_t, C_t^{(H)} \right) \right] \qquad (1)$$

#### 3.2.2 FINE-TUNING PDT

Similar to the fine-tuning strategy of Devlin et al. (2018), we simply fine-tune all PDT parameters on the downstream reward-annotated data. Since we have access to reward signals, we can now include them in the input sequence:

$$\underbrace{[s_1], [a_1], [\hat{R}_1],}_{\text{Pos.1}} \underbrace{[s_2], [a_2], [\hat{R}_2],}_{\text{Pos.2}} ..., \underbrace{[s_T], [a_T], [\hat{R}_T]}_{\text{Pos.}T}$$

During fine-tuning, instead of just predicting actions conditioned on the current state and context, i.e. $p_\theta\left(a_t|s_t, C_t^{(H)}\right)$, we also predict the RTGs conditioned on the current state, action, and context at every step, i.e. $p_\theta\left(\hat{R}_t|s_t, a_t, C_t^{(H)}\right)$. We found that empirically, this helps the model infer the RTG distribution which can help with the quality of the fine-tuned policy.

Overall, the training process in this step is similar to the training in the Decision Transformer, except for the addition of reward-prediction loss term. The fine-tuning loss is modified to contain both the MLE term for actions and the prediction loss for rewards, as described in Eq. 2. Note that the historical context $C_t^{(H)}$ also contain the now unmasked reward-to-go tokens.

$$\mathcal{L}_{FT}(\theta) = -\mathbb{E}_{(s_t, a_t, C_t, \hat{R}_t) \sim \mathcal{D}^{FT}}\left[\log p_\theta\left(a_t|s_t, C_t^{(H)}\right) + \log p_\theta\left(\hat{R}_t|s_t, a_t, C_t^{(H)}\right)\right] \tag{2}$$

**Return Prediction.** One limitation of the reward-to-go-conditioned structure of Decision Transformer is that each environment and each task requires a hand-picked initial reward-to-go to feed in. This can be un-intuitive and difficult: picking the best initial value can require prior knowledge about the training dataset and what rewards are feasibly achievable. Additionally, a fixed initial reward-to-go does not work well in environments that have random resets, such as D4RL's (Fu et al., 2020) pointmaze environment, since the distances to goal are different based on the randomized initial locations. In such environments, the best feasible reward-to-go depends on the initial state, and cannot be a fixed value. To overcome this while still learning a reward-to-go-conditioned actor, we learn to predict the best feasible reward-to-go based on the context and state. The model is trained to predict a probability distribution over the possible RTG tokens from the training dataset. At evaluation time, we pick the next RTG by sampling $N$ RTGs (e.g. $N = 250$) from the outputted probability distribution and picking the one with the highest value.

$$\hat{R}_t = \max(\hat{R}_t^1, \hat{R}_t^2 \ldots \hat{R}_t^N), \hat{R}_t^i \sim p_\theta(\hat{R}_t|s_t, a_t, C_t^{(H)}) \forall i \in (1, \ldots N) \tag{3}$$

One detail necessary for return prediction in continuous-reward environments (e.g. locomation mujoco tasks) is that we discretize the reward-to-gos into bins in order to learn a probability distribution over discerete values of RTGs rather than predicting scalar real-value returns.

### 3.2.3 EVALUATION

To rollout a trajectory using the model trained from PDT, we model it as a sequence of state-action-rewards. We alternate between two prediction phases for each time-step in the environment: (1) Using the context $(s_{t-H}, a_{t-H}, \hat{R}_{t-H}, ...s_t)$, predict the return-to-go $\hat{R}_t$ (see: Eq. 3). (2) Using the updated context $(s_{t-H}, a_{t-H}, \hat{R}_{t-H}, ...s_t, \hat{R}_t)$ predict the action $a_t$. In the first $H$ steps of the trajectory when there is not enough context, we use zero padding.

## 4 EXPERIMENTS

In the experiments we are interested in answering the following questions: (i) Can PDT be efficiently fine-tuned on downstream offline datasets where reward-labeled data is limited? (ii) How does PDT compare to

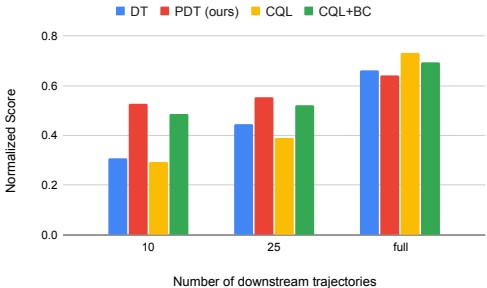

Figure 2: Aggregate results of the normalized scores across all domains and datasets. PDT achieves leading aggregate scores in the low-label regimes and is competetive with BC regularized CQL despite not learning a value funcction.

TD learning offline RL algorithms, despite not fitting any explicit value function? (iii) What are the most important design choices for adapting DTs to the semi-supervised offline RL setting?

## 4.1 ENVIRONMENTS AND DATASETS

We evaluate the performance of PDT on three simulated mujoco (HalfCheetah, Walker2d, and Hopper) and one robotic manipulation environment (FrankaKitchen) from the D4RL benchmark.

### 4.1.1 DATASETS

**Mujoco.** For the mujoco locomotion datasets, we consider three simulated locomotion agents and three datasets per agents (medium, medium-replay, and medium-expert), summing to a total of nine datasets. The datasets are collected in the following ways:

- The **medium** dataset is collected with partially trained SAC agent.
- The **medium-replay** dataset is the entire replay buffer of a SAC agent throughout training.
- The **medium-expert** is a mix between trajectories from the medium dataset and an expert policy.

**FrankaKitchen.** This environment consists of a 7-DOF kitchen robotic arm that is tasked to manipulate 4 objects in a particular order. We use *kicthen-partial-v0* which is collected with tele-operation and includes manipulating all objects, present in the environment, in different orders. However, each trajectory in the dataset is reward-labeled according to the order of the target task.

For pre-trained methods, we use the entire dataset in the D4RL benchmark to learn the behavioral prior. For fune-tuning we evaluate on different number of trajectories sampled from the dataset. In our experiments, **full** refers to using the entire reward-labeled dataset, **25** refers to using 25 randomly sampled trajectories, and **10** refers to using 10 randomly sampled trajectories. By doing so, we can investigate whether PDT can efficiently fine-tune the downstream offline tasks with small reward-labeled offline datasets?

### 4.1.2 EVALUATION PROTOCOL

For evaluating each of the results, we calculate each algorithm's performance by taking the average returns of the last 5 epochs for training on the downstream dataset. For each epoch, we evaluate 10 trajectory rollouts in the environment and report the average return based on the downstream task reward function.

We use the following approaches for comparison:

Table 1: Normalized scores for each dataset and each algorithm. PDT results are competative with other more sophisticated TD-learning methods that leverage the reward-free as well as the reward-annotated datasets. (i.e. CQL+BC)

| Domain | Dataset | Num trajectories | BC | DT | PDT (ours) | CQL | CQL+BC |
|---|---|---|---|---|---|---|---|
| Halfcheetah | medium-replay-v2 | 202 | | 0.82 | 0.79 ± 0.1 | **1.05** | 1.02 |
| | | 25 | 0.85 | 0.54 | 0.80 ± 0.06 | 0.80 | **0.93** |
| | | 10 | | 0.13 | 0.55 ± 0.05 | -0.09 | **0.82** |
| | medium-v2 | 1000 | | 0.94 | 0.92 ± 0.04.93 | 1.00 | **1.00** |
| | | 25 | 0.93 | 0.89 | 0.91 ± 0.04 | 0.92 | **0.96** |
| | | 10 | | 0.84 | **0.93 ± 0.032** | 0.83 | **0.92** |
| | medium-expert-v2 | 2000 | | 0.74 | **0.75 ± 0.067** | 0.26 | 0.50 |
| | | 25 | 0.63 | 0.41 | **0.75 ± 0.0769** | 0.27 | 0.42 |
| | | 10 | | 0.33 | **0.63 ± 0.077** | 0.34 | 0.43 |
| Hopper | medium-replay-v2 | 2039 | | 0.50 | 0.34 ± 0.025 | **0.80** | 0.55 |
| | | 25 | 0.24 | 0.20 | **0.17 ± 0.0230** | 0.03 | 0.12 |
| | | 10 | | **0.21** | 0.02 ± 0.027 | 0.09 | 0.17 |
| | medium-v2 | 2187 | | 0.54 | 0.55 ± 0.053 | **0.64** | 0.61 |
| | | 25 | 0.43 | 0.44 | 0.36 ± 0.08 | 0.54 | **0.57** |
| | | 10 | | 0.40 | 0.56 ± 0.042 | **0.57** | 0.56 |
| | medium-expert-v2 | 3214 | | 0.44 | 0.59 ± 0.0761 | 0.59 | **0.62** |
| | | 25 | 0.40 | 0.32 | 0.45 ± 0.03 | 0.38 | **0.50** |
| | | 10 | | 0.32 | **0.48 ±0.06** | 0.41 | 0.28 |
| Walker2d | medium-replay-v2 | 1093 | | 0.48 | 0.13 ± 0.13 | **0.80** | 0.76 |
| | | 25 | 0.69 | **0.17** | 0.15 ± 0.09 | 0.05 | 0.04 |
| | | 10 | | 0.08 | **0.20 ± 0.05** | 0.03 | 0.02 |
| | medium-v2 | 1191 | | 0.80 | 0.78 ± 0.06 | **0.83** | 0.80 |
| | | 25 | 0.70 | 0.56 | 0.77 ± 0.05 | 0.04 | **0.80** |
| | | 10 | | 0.30 | 0.44 ± 0.09 | 0.02 | **0.80** |
| | medium-expert-v2 | 2191 | | 0.97 | **0.99 ± 0.0** | 0.85 | 0.92 |
| | | 25 | 0.89 | 0.62 | **0.98 ± 0.01** | 0.63 | 0.86 |
| | | 10 | | 0.22 | **0.97 ± 0.02** | 0.48 | 0.75 |
| FrankaKitchen | kitchen-partial-v0 | 601 | | 0.39 | **0.57 ± 0.06** | 0.50 | 0.17 |
| | | 25 | 0.00 | **0.31** | 0.19 ± 0.07 | 0.25 | 0.00 |
| | | 10 | | 0.25 | **0.51 ± 0.03** | 0.25 | 0.10 |
| Average | | full | | 0.66 | 0.64 | **0.73** | 0.69 |
| | | 25 | 0.58 | 0.44 | **0.55** | 0.39 | 0.52 |
| | | 10 | | 0.31 | **0.53** | 0.29 | 0.49 |

- **PDT (ours):** Our approach for pre-training DT on reward-free data and fine-tuning on the reward-annotated dataset.

- **DT:** Decision Transformer which uses transformers to auto-regressively predict actions conditioned on current state, reward-to-go, and context.

- **BC:** This baseline is simply a behavioral cloning on the reward-free dataset. Since the trajectories in these datasets are not entirely collected from experts, we do not expect this baseline to be superior and be able to achieve beyond the average of the returns in the dataset.

- **CQL:** Conservative Q-Learning (Kumar et al., 2020), which is a popular model-free offline RL algorithm. This baseline is only trained on the downstream data but uses the reward annotations.

- **CQL+BC:** BC regularized CQL. This is an extension of CQL that also regularizes the policy's distribution to match that of the pre-training dataset. We simply apply CQL on the downstream dataset while regularizing the policy to imitate the actions in the pre-training dataset with an additional loss term to the policy updates.

We train BC on the entire reward-free dataset; CQL, and DT only on the reward-annotated dataset, and CQL+BC on both reward-free and reward-annotated dataset. CQL+BC, similar to PDT, leverages both the reward-free and reward-annotated datasets.

**Results.** Figure 2 shows the aggregate of the normalized returns across all tasks as a function of the number of reward-labeled fine-tuning trajectories. As we can see, compared to DT and CQL, the algorithms that leverage reward-free datasets (i.e. PDT and CQL+BC) perform significantly better in low data regimes. It is also noted that PDT's performance is competitive with CQL+BC, but the simplicity and flexibility of its architecture makes PDT a suitable candidate for wide-spread adoption in semi-supervised offline RL settings.

To illustrate the granular results on each environment, we have included Table 1. The scores are normalized to the maximal return of the trajectories in each dataset. As we can see, even though both DT and CQL mostly achieve good results in full data regimes, their performance significantly drops in settings where labeled-data is limited. In the kitchen environment in particular, CQL+BC completely collapses as the desired multi-modal behavior of the agent cannot be captured by a simple BC regularization, while PDT can recover the task-specific behavior in low data regimes.

## 4.2 ABLATION STUDIES

In this section we study different components of the PDT algorithm to provide insight on the contribution of each part. In particular, we are interested in performing the following ablations:

**The effect of reward prediction during fine-tuning.** During the fine-tuning step of our framework we not only predict the next actions conditioned on the current state, reward-to-go, and context, but we also predict the distribution of possible remaining reward-to-gos. To this end, we study the effect of this component by comparing to a case where it is not used. We run an experiment with the *maze2d-large-v1* D4RL in which the agent needs to know how far it is from the goal location to be able to navigate to it properly.

| # Trajectories | PDT | PDT w/o rew. | DT |
|---|---|---|---|
| 10 | **0.69** | 0.61 | 0.56 |
| 20 | **0.71** | 0.61 | 0.40 |
| 50 | **0.69** | 0.67 | 0.60 |
| 100 | **0.72** | 0.70 | 0.55 |

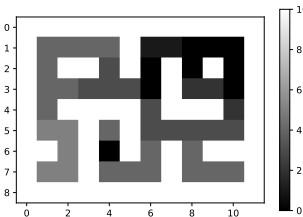

Figure 3: Left: Ablation on effect of reward prediction on pointmass maze task, Right: Prediction of returns-to-go of different discrete states in the PointMaze environment

In this experiment, we use our model to output the best possible return-to-go given different starting locations in the maze. In Figure 3 (right) a darker square indicates a bin that represents a better return-to-go. This means that if the point mass starts in that state, it will be able to quickly reach the goal square of (9, 1). As expected, states closer to (9, 1) have a better reward-to-go.

**The effect of masking reward token attention during pre-training.** During pre-training we mask the reward attention to force the model to not attend to reward tokens, and to auto-regressively model the next actions conditioned on the current state, and context without rewards. Here, we compare the post fine-tuning performance of PDT with a version that does not apply attention masks during pre-training. We show three experiments on the medium-expert suite in mujoco, since the effect is seen most clearly in this dataset type. Including masking during pre-training, improves results regardless of how many trajectories the fine-tuning dataset includes.

**Comparison to other choices of fine-tuning.** PDT's strategy for fine-tuning is the most straight-forward instantiation of this idea which is fine-tuning *all* pre-trained parameters. Here, we compare this particular design choice to other common fine-tuning patterns seen in transformer literature. In particular we compare,

Table 2: Ablation on effect of reward masking during pre-training

| task | number of trajectories | PDT (w/ masking) | PDT (wo/ masking) |
|---|---|---|---|
| hopper-medium-expert-v2 | 3214 | 0.59 | 0.47 |
| | 25 | 0.45 | 0.45 |
| | 10 | 0.48 | 0.38 |
| halfcheetah-medium-expert-v2 | 2000 | 0.75 | 0.78 |
| | 25 | 0.75 | 0.71 |
| | 10 | 0.63 | 0.59 |
| Walker2d-medium-expert-v2 | 2191 | 0.99 | 0.97 |
| | 25 | 0.98 | 0.98 |
| | 10 | 0.97 | 0.94 |

fine-tuning with frozen self-attention layer and fine-tuning with both frozen self-attention and feed-forward layers. The later choice was proposed recently on Lu et al. (2021).

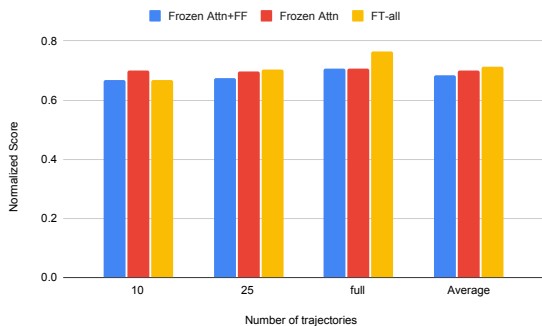

Figure 4: Ablation on the effect of which PDT parameters to fine-tune. Numbers presented show the average normalized score over all medium and medium-expert mujoco experiments presented in our main results.

Empirically, we see that fine-tuning all of the parameters, on average, outperforms fine-tuning only a subset of the parameters, although the difference overall is not extreme. Interestingly, in the very-low data domain of 10 trajectories, it does appear that only freezing the attention parameters does lead to the best performance, but for 25 trajectories and the full trajectories, fine-tuning all parameters does best.

## 5  CONCLUSION

We presented PDT, a simple and effective architecture amenable to semi-supervised offline RL settings where the architecture is pre-trained on reward-free and task-agnostic datasets and then fine-tuned to a task-specific downstream dataset. Our technical contribution is extending Decision Transformers to this setting by masking reward tokens during pre-training and predicting remaining reward-to-gos during fine-tuning. We show how PDT can outperform existing offline RL methods in low data regimes. The simple training procedure of PDT makes it a great candidate for wide-spread adoption in semi-supervised offline RL settings.

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
