# OpenReview forum: "Semi-supervised Offline Reinforcement Learning with Pre-trained Decision Transformers"
_ICLR.cc/2022/Conference — ICLR 2022 Submitted_

### Official Review · Reviewer_z2Wf · 2021-10-31

**Correctness:** 3
**Technical Novelty And Significance:** 2
**Empirical Novelty And Significance:** 2
**Recommendation:** 5
**Confidence:** 4

**Main Review:**

Strengths:
1.	The semi-supervised setting seems new in offline reinforcement learning.
2.	The proposed method is simple.
3.	Ablations have been provided to study the effects of reward prediction during fine-tuning, masking reward token attention during pre-training, and other choices of fine-tuning.

Weakness:
1.	The importance of fine-tuning is not well justified. The pre-trained decision transformer model has already achieved sufficiently good performance without reward annotations. Why do we need to fine-tune the pretrained model on a subset of the pretrained dataset with reward annotations? The significance of fine-tuning can be shown on more diverse and challenging datasets.
2.	Pre-training a decision transformer on a large dataset is expensive. Fine-tuning a full pre-trained decision transformer on downstream dataset is not efficient. It would be better if a light-weight fine-tuning method can be applied.
3.	Despite the authors provided ablations, the discussion and conclusion on which factor is most effective is sufficient. For example, Why PDT with masking reward token during pre-training works slightly better than PDT without masking?

Other questions:
1.	Table 1 shows the fine-tuning results of PDT. How does the pre-trained decision transformer perform? Is it better than the reported PDT results? If the pretrained decision transformer outperform the PDT fine-tuned on smaller downstream datasets, then the fine-tuning is not that significant.
2.	Some important experimental details are missing. For example, how many epochs were used for fine-tuning and how many steps in each epoch? If the fine-tuning is expensive, then why not train the DT on the small dataset with reward annotations from scratch?


**Summary Of The Paper:**

This paper presents Pre-trained Decision Transformer (PDT) for semi-supervised offline reinforcement learning. PDT first pre-train a decision transformer model on the trajectory dataset without rewards, and then fine-tune on a smaller dataset with reward annotations. Empirically, PDT achieves compatible performance with offline reinforcement learning baselines.

**Summary Of The Review:**

Overall, I vote for weak rejection. Despite its simplicity, the novelty and significance of the proposed method are limited. My major concern is the importance of fine-tuning is not well supported in the experiments. It can be enhanced by showing the significance of fine-tuning on more diverse downstream datasets that collected from related but more challenging tasks. Hopefully the authors can address my concern in the rebuttal period.

---

### Official Review · Reviewer_qGs8 · 2021-11-01

**Correctness:** 3
**Technical Novelty And Significance:** 2
**Empirical Novelty And Significance:** Not applicable
**Recommendation:** 5
**Confidence:** 4

**Main Review:**

== Strengths ==

- Transfer learning like pre-train & fine-tune and self-supervised in deep reinforcement learning is an important and interesting problem.
- PDT can leverage reward-free trajectories and improve the agent in fine-tune stage with limited 'labeled data'. It could be a promising solution to the problem of data efficiency in deep RL.
- The proposed approach achieves improvements on D4RL datasets.

== Weaknesses ==

- I think the contribution and novelty of this work is not enough, the main difference from Decision Transformers is the MLE loss, which is minor, and the reward prediction. So the ablation study on effect of reward prediction during fine-tuning on mujoco or FrankaKitchen in D4RL is also important.
- As an improved version of decision transformers, I noticed that the order of trajectory representation (s_t, a_t, \hat{R}_t} is different from that in DT (\hat{R}_t, s_t, a_t). As a results, in Section 3.1, \hat{R}_t is not included in the C_t^{(H)}, which may be different from the DT version. There should be an explanation for this change.
- In Figure 1(Right), the \hat{R}_{t-1} is both of the output and input of transformer, so the model is using the predicted RTG as input? Why not use the real rewards from environment at evaluation time?
- 'For pre-trained methods, we use the entire dataset in the D4RL benchmark to learn the behavioral prior.', so the entire dataset means all trajectories for one task or all tasks? As listed in contributions, 'PDT can efficiently be fine-tuned for a range of diverse downstream tasks using the same pre-trained model', how different tasks with different sizes of states and actions share the same embedding layers? I think more details of pre-train stage are required.
- As for the experiments results, I noticed that in some cases, when there are 25 trajectories, the performance is worse than 10 and full, and the gap is quite big, what is the reason for this?

**Summary Of The Paper:**

This paper introduces a self-supervised approach in deep reinforcement learning with decision transformers, which is widely used in CV and NLP tasks. The algorithm can achieve better performace than traditional methods like DT, BC and CQL, especially when data is insufficient.

**Summary Of The Review:**

The paper introduces a self-supervised approach to improve the performance of decision transformers in deep RL when data is limited. But the contribution and novelty is not enough. So I give the score of 5.

---

### Official Review · Reviewer_fBou · 2021-11-05

**Correctness:** 2
**Technical Novelty And Significance:** 2
**Empirical Novelty And Significance:** 2
**Recommendation:** 3
**Confidence:** 3

**Main Review:**

### Key Strengths
  1. I think this is a good problem area -- improving efficiency of RL systems and providing a vehicle for generalizing RL knowledge across settings seems very valuable.
  2. I think the method presented here is intuitive; withholding reward information while giving the model a better ability to model action sequences seems like it ought to help.

### Key Weaknesses
  1. I'm confused by something key in your results. Firstly, what does the "-v#" suffix indicate in your data tables? You don't make that clear anywhere as far as I can tell. This is relevant because your performance #s don't appear to match those in the decision transformer paper (for the DT or CQL runs). Why are these different?
  2. I'm not as convinced by your results. In the context of comparing across a wide variety of tasks, I think using a metric like the # of wins or average rank is superior to average performance across tasks. In your setting, vs. CQL+BD on FULL you win 3 / 10 times, on 25 you win 5 / 10 times, and on 10 you win 5 / 10 times with 2 / 10 ties. In terms of overall rank, you take first 3/10, 3/10, and 5/10 (with 1/10 ties) times for full, 25, and 10 respectively. These don't suggest to me that you beat CQL+BD or the overall suite of baselines in a compelling fashion. I also think it is a problem if your method does worse than existing approaches in the full-data setting. At the very least, it should be matching those approaches, otherwise why would I use it in real-world settings? If it will only over advantages in a narrow range of # of observed trajectoreis (where the # allowed likely varies by setting/task), using it could risk yielding worse test set performance vs. standard methods.
     A related point to this is that dropping the uncertainty in your average performance cell in table 1 isn't acceptable -- you have all the variances for the PDT model listed in the individual cells above; propagate those through the averaging operation so we can see to what extent things are meaningfully separated. Similarly, there need to be error bars in Figure 2. I am definitely concerned that in some cases (e.g., 25 and 10) the propagated uncertainty will be so large that even in the average comparison there isn't clear separation between your method and CQL+BC.
    Note that this concern is what motivates the score of "2" for the correctness column. If the presentation of results could be cleaned up and a more thorough comparison across methods performed, I'd improve that numerical score.
  3. While I don't know of any works that do this exactly, I do think there is only limited novelty/technical sophistication here. It seems to me that you have largely just split the training phase of the DT into a PT piece where rewards were occluded then a FT piece with rewards present. While that contribution does have merit, I'm not sure it is sufficient for an ICLR submission.

### Minor Weaknesses:
  1. In your setting, what are the pre-training datasets vs. the fine-tuning datasets? Or are they the same and only the reward values are occluded during PT? An important part of PT/FT in the general domain is the inherent question of domain generalizability in going from a PT dataset to a FT dataset, which come from different distributions. It seems like this may be missing from your setting? If I am correct and there is no question of domain generalizability / task transfer, then I feel your paper is slightly misrepresenting its problem. Rather than doing PT/FT, this is really more like classical semi-supervised learning, where you have instances of labels (or in this case, rewards) being missing at random from your training data. This isn't really a big problem, as you compare to semi-supervised methods as appropriate, but re-framing it to be more in line with that perspective would help make the paper tighter.


**Summary Of The Paper:**

### What is the Problem / Question?
Pre-training has not been thoroughly explored within RL, so it is unknown how effectively task-free unlabeled datasets can be used for PT in an RL context. The authors present a simple strategy for such PT for RL that shows benefits with limited reward annotations.

### Why is it impactful?
RL is a famously expensive discipline of ML which nonetheless has huge application areas and potential in theoretial and deployment contexts. Being able to translate the improvements offerred by pre-training to the RL context would help empower this already important technique significantly.

### Why is it hard? Why have previous approaches failed?
There are several previous approaches to this problem, including some the authors readily identify in their study, such as the CQL+BC method which uses unlabeled data to perform semi-supervised offline RL by adapting policy prior data via unlabeled data.

### How do they solve it?
The authors propose using the (already published) decision transformer model, but pre-training that model first on a dataset with only action sequences, without any reward information.

### How do they validate their solution?
The authors compare their proposed framework against existing models, including both the traditional DT, a BC baseline, and the CQL and CQL+BC methods (the latter of which is a semi-supervised approach). They find slight improvements over existing methods in the extreme few-shot settings (10 observed trajectories at FT time), with further reduced gains at larger #s of observed trajectories, culminating in worse performance than traditional methods at full data scale.


**Summary Of The Review:**

I think this is a great start, but insufficient for ICLR, so I'm recommending reject here. This is motivated by two key problems: (1) The results don't convince me there is real improvement here, in particular vs. the CQL+BC baseline, and I do think the degradation in performance at the full setting is a major issue, and (2) I'm not convinced there is sufficient technical novelty / sophistication here. This submission in its current form (modulo some edits to improve the presentaition of the results and more accurately reflect the performance relationship between the various methods) might be a great fit for a workshop, and improvements to the PT/FT method to increase significance of results and/or expand the technical sophistication of the method would make it a better fit for ICLR.

---

### Official Review · Reviewer_xxc6 · 2021-11-05

**Correctness:** 3
**Technical Novelty And Significance:** 2
**Empirical Novelty And Significance:** 2
**Recommendation:** 3
**Confidence:** 4

**Main Review:**

Extending decision transformer to work on unlabelled datasets is an interesting research direction, one can imagine a large scale DT trained on a large-scale unlabelled dataset can enable efficient transfer to unseen tasks and envs in a similar nature to GPT-3. The specific setting that this paper operates in though is quite odd.

The paper proposes a way to pre-train Decision Transformers on unlabelled  or reward-free data, but the experiments are conducted on data collected by expert/semi-expert agents which did use reward during their training. This also means that the offline datasets are biased towards high reward trajectories. So, calling the pre-training setup “reward-free” is misleading. There could be several alternative experimental setups which can make more realistic reward-free setups:

  * Pre-training on data collected by a random policy
  * Pre-training on data collected by an exploratory policy which doesn’t use extrinsic reward (like curiosity etc.)
  * Pre-training on data collected from a different task or reward function. (Task-transfer)
  * Pre-training on in-the-wild videos.

On experimental results, PDT doesn’t show much benefit over simple BC or CQL+BC.

The most aspects of the paper are on how to improve fine-tuning performance, and which design choices matter. I would recommend authors to use a more appropriate experimental setup, or show results for transfer, which would make a much more compelling case.

**Summary Of The Paper:**

This paper proposes pre-trained decision transformers (PDT) that are trained on an offline dataset collected by an expert/semi-expert agent with reward information masked out. PDT is then evaluated by fine-tuning on the same task.

**Summary Of The Review:**

The paper has major shortcomings with the experimental setup considered to evaluate the efficacy of pre-trained decision transformers, and falls short of outperforming simple baselines. For these reasons, I recommend rejection at this point.

---

### Decision · Program_Chairs · 2022-01-20

**Decision:**

Reject

**Comment:**

This paper explored pre-training for deep offline reinforcement learning, developing a method that first pre-trained decision transformers on trajectories without rewards, and then fine-tuned on limited data with rewards. The reviewers were pleased with the overall research questions and directions, but found that they were substantial shortcomings in the experimental setup and results that make this paper not yet suitable for inclusion. The approach is relatively simple and straightforward, which is actually a good thing, but that means that it must be correspondingly investigated and developed with convincing empirical results. Unfortunately, there are a number of open questions about the experimental set up, and the results are not convincing that the method is effective against alternatives, as detailed in the reviews. There was no author rebuttal.